# The Carbon Source Controls the Secretion and Yield of Polysaccharide-Hydrolyzing Enzymes of Basidiomycetes

**DOI:** 10.3390/biom11091341

**Published:** 2021-09-10

**Authors:** Eka Metreveli, Tamar Khardziani, Vladimir Elisashvili

**Affiliations:** Institute of Microbial Biotechnology, Agricultural University of Georgia, 0131 Tbilisi, Georgia; e.metreveli@agruni.edu.ge (E.M.); t.khardziani@agruni.edu.ge (T.K.)

**Keywords:** wood-rotting basidiomycetes, carbon source, cellulase, regulation of synthesis, saccharification

## Abstract

In the present study, the polysaccharide-hydrolyzing secretomes of *Irpex lacteus* (Fr.) Fr. (1828) BCC104, *Pycnoporus coccineus* (Fr.) Bondartsev and Singer (1941) BCC310, and *Schizophyllum commune* Fr. (1815) BCC632 were analyzed in submerged fermentation conditions to elucidate the effect of chemically and structurally different carbon sources on the expression of cellulases and xylanase. Among polymeric substrates, crystalline cellulose appeared to be the best carbon source providing the highest endoglucanase, total cellulase, and xylanase activities. Mandarin pomace as a growth substrate for *S. commune* allowed to achieve comparatively high volumetric activities of all target enzymes while wheat straw induced a significant secretion of cellulase and xylanase activities of *I. lacteus* and *P. coccineus*. An additive effect on the secretion of cellulases and xylanases by the tested fungi was observed when crystalline cellulose was combined with mandarin pomace. In *I. lacteus* the cellulase and xylanase production is inducible in the presence of cellulose-rich substrates but is suppressed in the presence of an excess of easily metabolizable carbon source. These enzymes are expressed in a coordinated manner under all conditions studied. It was shown that the substitution of glucose in the inoculum medium with Avicel provides accelerated enzyme production by *I. lacteus* and higher cellulase and xylanase activities of the fungus. These results add new knowledge to the physiology of basidiomycetes to improve cellulase production.

## 1. Introduction

Plant biomass is abundant, renewable, rich in carbohydrates, and the most suitable resource for large-scale production of bioenergy and various organic chemicals [1,2]. It is a cheap but complex material consisting of polymers of cellulose, hemicellulose, and lignin; therefore, a wide range of hydrolytic and oxidative enzymes is required for their degradation. Hydrolysis of biomass polysaccharides into fermentable sugars by cellulases and hemicellulases is the key step for enzymatic conversion of lignocellulose [1,2,3,4]. Cellulases comprise endoglucanases (EC 3.2.1.4) which cleave internal β-1,4-glucosidic bonds of cellulose chains, exoglucanases (EC 3.2.1.91) which processively act on the reducing and non-reducing ends of cellulose to release short-chain cello-oligosaccharides, and β-glucosidases (EC 3.2.1.21) which hydrolyze soluble cello-oligosaccharides to glucose. As far as hemicellulases are concerned, endo-β-1,4-xylanases (EC 3.2.1.8) and β-xylosidases (EC 3.2.1.37) and auxiliary enzymes are required for their complete hydrolysis.

Cellulases are widely used in biofuel, food and feed, textile, paper-pulp, cosmetics, chemicals, and other industries [1,2]. The main challenges for wide and large-scale application of cellulases and xylanases remain the reduction of their cost and the development of more efficient enzyme cocktails with high specific activity and stability [3,4]. Wood-decomposing white-rot basidiomycetes (WRB) are excellent producers of enzymes that deconstruct the cell wall of plants [5]. Due to their ecological and biological peculiarities, they are highly adapted to different environments and resources. Moreover, some of them have shown exceptional potential for the production of individual groups of hydrolytic enzymes under appropriate cultivation conditions. Thus, *Coprinellus disseminatus* produced 469 U/mL of alkali-thermotolerant xylanase along with negligible cellulase activity [6] while *Armillaria gemina* secreted up to 146 U endoglucanase/mL, 15 U β-glucosidase/mL, and 1.72 U FPA/mL [7]. Further, Jagtap et al. [8] achieved very high β-glucosidase activity (45.2 U/mL) in the submerged cultivation of *Pholiota adiposa* in a medium containing rice straw and corn steep powder. High cellulase and xylanase activities were revealed in submerged cultivation of *Irpex lacteus* and *Schizophyllum commune* [9,10]. However, unlike soft rot ascomycetes, such as *Trichoderma reesei* and *Aspergillus niger*, which are the most studied cellulolytic fungi and widely used in biorefinery microorganisms, the physiology of cellulose degradation and mechanism of cellulases synthesis regulation in WRB remains poorly understood. Therefore, more in-depth studies are required to understand how specific environmental factors modulate the secretion of individual cellulases and xylanase to develop the fungus enzyme system and technological process for their industrial application.

In the present work, profiles of the polysaccharide-hydrolyzing secretomes of three saprotrophic WRB species were analyzed in their submerged cultivation to elucidate the effect of chemically and structurally different carbon sources on the expression of cellulases and xylanase and to establish substrate specificities and enzyme production patterns of the tested fungi. The results obtained contribute to a better understanding of the effect of the type of lignocellulosic biomass and carbon source on the induction/repression of cellulolytic enzymes.

## 2. Materials and Methods

### 2.1. Organisms and Inoculum Preparation

*Irpex lacteus* (Fr.) Fr. (1828) BCC104, *Pycnoporus coccineus* (Fr.) Bondartsev and Singer (1941) BCC310, and *Schizophyllum commune* Fr. (1815) BCC632 isolated from tree branches in the forests of Georgia and deposited in the basidiomycetes culture collection of the Institute of Microbial Biotechnology have been used in this study. The fungal inocula were prepared by growing the mycelia on a rotary shaker (New Brunswick Scientific, Edison, NJ, USA) at 160 rpm and 27 °C in 250 mL flasks containing 100 mL of standard medium (SM) (g/L): glucose–15.0, KH_2_PO_4_–1.0, K_2_HPO_4_–0.2, MgSO_4_·7H_2_O–0.5, peptone–3.0, yeast extract–3.0, pH 6.0. After 7 days of fungal cultivation mycelial pellets were harvested and homogenized using a Waring laboratory blender (Waring Commercial, Torrington, CT, USA).

### 2.2. Cultivation Conditions

Submerged cultivation of fungi was carried out using rotary shakers Innova 44 (New Brunswick, NJ, USA) at 160 rpm and 27 °C in 250 mL flasks containing 100 mL of the medium of following composition (g/L): KH_2_PO_4_–1.0, K_2_HPO_4_–0.2, MgSO_4_·7H_2_O–0.5, peptone–7.0, yeast extract–5.0, pH 6.0. Crystalline cellulose at a concentration of 15 g/L and milled to powder wheat straw, beech sawdust, wheat bran, and mandarin squeeze obtained from the juice manufacturing company in Kobuleti were used at a concentration of 40 g/L as carbon sources. Moreover, the effect of adding glycerol to the 1.5% Avicel-containing medium on enzyme synthesis was assessed in special experiments.

During the fungus cultivation, at predetermined time intervals, 1–2 mL samples were taken from the flasks, the solids were separated by centrifugation at 10,000× *g* for 10 min at 4 °C and the supernatants were analyzed for pH, reducing sugars, and enzyme activities.

### 2.3. Cultivation in a Fermenter

To scale up the cellulase and xylanase production by *I. lacteus*, one cultivation of the fungus was performed in the 7 L fermenter LiFlus GX (Biotron, Incheon, Korea) equipped with pH, temperature, pO2 probes, and three Rushton impellers. The fermenter was filled with 5 L of medium containing per liter: 20 g Avicel, 7.5 g glycerol, 1 g KH_2_PO_4_, 0.5 g MgSO_4_, 7 g peptone, 7 g yeast extract. Polypropylene glycol 2000 (3 mL) was added as an antifoam agent and the medium pH was adjusted to 6.0. The fermenter was sterilized (121 °C, 40 min) and inoculated with 500 mL of homogenized mycelium grown in SM containing crystalline cellulose instead of glucose. Fermentation was performed with baffles at 27 °C and the constant airflow rate of 1 L/L/min. During the fermentation process, samples were collected daily and analyzed for enzyme activity. After 8 days of fermentation, the fungal biomass was separated from the culture liquid by filtration followed by centrifugation at 6000 rpm for 20 min at 4 °C. The enzyme preparation was isolated from the culture liquid by precipitation with ammonium sulfate at 70% saturation and the precipitate was dissolved in 0.05 M phosphate buffer (pH 6.0).

### 2.4. Wheat Straw Saccharification

Enzymatic hydrolysis of wheat straw was carried out under standard conditions (0.1 mM of citrate buffer, pH 5.0, 40 °C) with gentle agitation at 150 rpm for 24 h. Pretreated with 1.5% NaOH wheat straw contained 67.1% cellulose, 15.9% hemicellulose, and 7.3% lignin. Cellulase from *Aspergillus niger* (Sigma–Aldrich, Saint Louis, MO, USA) and crude enzyme preparations isolated (as described above) from the supernatant of *I. lacteus* culture in fermenter were used as enzyme sources. The reaction mixture (10 mL) contained 100, 200, and 400 mg substrate and 20 filter paper units (FPU)/g substrate. Samples were taken from the reaction mixtures after 0, 3, 6, and 24 h of saccharification, heated in a boiling water bath for 2 min, and then centrifuged at 10,000× *g* for 5 min at 4 °C. The supernatants were analyzed for reducing sugars using the dinitrosalicylic acid reagent method [11].

### 2.5. Analytical Methods

The total cellulase activity (filter paper activity, FPA) was measured with Whatman filter paper No. 1 according to IUPAC recommendations [12]. The reaction mixture containing a 50 mg string of filter paper Whatman No. 1 (Whatman Internationl, Maidstone, UK), 0.8 mL of a 50 mM citrate buffer (pH 5.0), and 0.2 mL appropriately diluted supernatant was incubated at 50 °C for 60 min. Endoglucanase (CMCase) activity was assayed by mixing 70 µL appropriately diluted samples with 630 µL of 1% low-viscosity carboxymethyl cellulose in 50 mM citrate buffer (pH 5.0) at 50 °C for five minutes [12]. Xylanase activity was determined at the same conditions using 1% birchwood xylan (Roth 7500) in 50 mM citrate buffer (pH 5.0) at 50 °C for 10 min [13]. Glucose and xylose standard curves were used to calculate the cellulase and xylanase activities. In all assays, the release of reducing sugars was measured using the dinitrosalicylic acid reagent method [11]. One unit of enzyme activity was defined as the amount of enzyme, releasing 1 μmol of reducing sugars per minute. To measure β-glucosidase and β-xylosidase activities, the reaction mixture containing 1.8 mL of 2 mM solutions of *p*-nitrophenyl-β-d-glucopyranoside or *p*-nitrophenyl-β-d-xylopyranoside in 0.05 M acetate buffer, pH 4.8, and 0.2 mL of the enzyme solution was incubated at 50 °C for 10 min [14]. One unit of enzyme activity was defined as the amount of enzyme releasing 1 μmol of *p*-nitrophenol per minute.

### 2.6. Statistical Analysis

All experiments were performed twice using three replicates each time. The results are expressed as the mean ± SD. The mean values, as well as standard deviations, were calculated by the Excel program (Microsoft Office 2010 package).

## 3. Results and Discussion

### 3.1. Effect of the Polymeric Carbon Sources on Basidiomycetes Enzyme Activity

Chemically and structurally different crystalline cellulose, wheat straw, and mandarin pomace were selected as carbon sources and potential stimulators of cellulase and xylanase activities production by the three fungi. Due to the high content of sugars and organic acids, mandarin pomace ensured rapid and abundant growth of all fungi, while wheat straw showed the weakest growth. The fungi secreted large amounts of cellulase and xylanase activities regardless of the material tested; however, enzyme yields varied significantly. Among the three substrates tested, crystalline cellulose was found to be the best carbon source providing the highest cellulase and xylanase activity of all fungi (Table 1). This finding is in agreement with several reports showing cellulases and xylanases production by *Basidiomycota* species in the presence of several potential inducers [15,16,17].

The use of mandarin pomace as a growth substrate for *S. commune* allowed us to achieve comparatively high volumetric activities of all target enzymes. At the same time, the lignified material, wheat straw, induced a significant secretion of cellulase and xylanase activities of *I. lacteus* and *P. coccineus* but turned out to be a poor substrate for the production of these enzymes by *S. commune*. It should be noted that during the fermentation of straw and mandarin pomace the pH of the media was significantly higher than those in the media with Avicel, especially in the second half of the WRB cultivation. It is possible that, in this case, the hydrolysis of polysaccharides and the supply of fungi with carbon sources were limited, since the maximum catalytic activity of cellulases and xylanases of basidiomycetes is usually observed at pH 5.

To ensure rapid and abundant initial growth of fungi, Avicel and wheat straw-containing media were supplemented with mandarin pomace. Undoubtedly, the introduction of additional nutrition in the form of mandarin pomace favored an increase in the biomass of fungi and, accordingly, enzymatic activity. Nevertheless, the results presented in Table 1, in several variants of the experiment, clearly show the additive effect of the two substrates on the secretion of enzymes. In particular, an almost twofold increase in CMCase activity of *S. commune* and *I. lacteus* was recorded when the wheat straw-based medium was supplemented with mandarin pomace. Moreover, the xylanase activity of *S. commune* was increased by 4 times. Further, the presence of mandarin pomace in the Avicel-containing medium favored the accumulation of β-glucosidase and β-xylosidase by the tested fungi.

Analysis of the profiles of enzyme production showed that they depend both on the type of fungus and the type of growth substrate. In Avicel containing cultures CMCase, xylanase, and FP activities of the fungi gradually increased and achieved maximum after 11–14 days of cultivation, while in mandarin pomace or wheat straw containing media enzyme activity in individual fungal cultures peaked after 8 days of cultivation followed by a steady decline in the activity thereafter. It is worth noting that in the cultures of *P. coccineus* and *I. lacteus* β-glucosidase and β-xylosidase activities achieved their maximum earlier than the polysaccharide-hydrolyzing enzymes.

Among the fungi studied *S. commune* appeared to be an outstanding producer of xylanase and β-glucosidase (740 and 18.6 U/mL, respectively) whereas *P. coccineus* and *I. lacteus* were distinguished by remarkable cellulase activity (Table 1). The highest endoglucanase activity (82 U/mL) was revealed in the cultivation of *P. coccineus* followed by *I. lacteus* in a medium containing both Avicel and mandarin pomace as growth substrates. In the same medium, *I. lacteus* secreted the highest FPA (6.8 U/mL). For further study, *I. lacteus* was chosen not only because of the high activity of the enzymes involved in the hydrolysis of polysaccharides. This species is characterized by a high potential for colonization of a wide variety of lignocellulosic materials due to its ability to produce all the enzymes necessary for the decomposition of polymers of plant raw materials [18,19].

### 3.2. Effect of Glycerol as an Additional Easily Metabolizable Carbon Source

In a subsequent set of experiments, the effect of glycerol as an additional easily metabolized carbon source on the production of hydrolases during submerged fermentation of Avicel and wheat straw by *I. lacteus* was studied. There were several reasons for this. Firstly, microcrystalline cellulose and lignified straw are difficult to decompose growth substrates that slow down the initial development of fungal culture. At the same time, we observed that the use of mandarin pomace, which is characterized by a high content of sugars and organic acids, accelerates the growth of fungi and the rapid accumulation of biomass, which promotes the accumulation of enzymes. Secondly, the use of a high concentration of mandarin pomace in the nutrient medium complicates the subsequent purification of the enzyme. Thirdly, glycerol is a cheap byproduct in the biodiesel manufacturing process and an excellent carbon source for the cultivation of majority basidiomycetes. Finally, it was important to elucidate the features of the synthesis of cellulases by *I. lacteus* in the presence of a readily metabolizable carbon source.

The results presented in Figure 1A show that when the fungus was cultivated in a medium with crystalline cellulose, sufficiently high activity of cellulase was already detected after two days to provide the culture with a source of carbon and energy. Then the activity of the secreted enzyme rapidly increased, reaching a maximum on the tenth day of cultivation of the fungus. After this, the activity of the extracellular enzyme decreased. When the Avicel-containing medium was supplemented with glycerol, no CMCase production was observed even though the inducer was also present. It is worth noting that the higher was the glycerol concentration in the medium the longer was the period of catabolite repression of cellulase synthesis. However, when the level of available glycerol decreased, obviously due to fungal metabolism, the production of enzyme was initiated even with a higher rate than in the control culture. Nevertheless, we believe that the accelerated rate of accumulation of the enzyme, as well as the significantly increased yields of the enzyme, are simply explained by the accumulation of greater biomass of the fungus in the presence of glycerol. Interestingly, the secretion of *I. lacteus* xylanase followed similar production profiles as CMCase (Figure 1B), and they appeared to be co-expressed. It should be noted that the cultivation of *I. lacteus* in a medium containing only glycerol accompanied by an abundant growth of fungal mycelium, but very low enzyme activity. If we consider this activity as the basal level of the enzyme, then the induction ratio for *I. lacteus* CMCase is 67, and for xylanase-22. Thus, it can be concluded that as in many other WRB [4,10,16] cellulase and xylanase of *I. lacteus* are inducible enzymes only in the presence of cellulose-containing materials and another mechanism of cell economy, namely, catabolite repression of the cellulases and xylanases synthesis, by easily metabolizable carbon sources is inherent in this fungus. 

### 3.3. Effect of the Type of Carbon Source in the Inoculum Preparation Medium

Recently, we showed that preliminary adaptation to the substrate in the inoculation medium may be an important factor determining the biosynthetic activity of some fungal cultures [20]. In this study, we compared the enzymatic activity of *I. lacteus*, an inoculum of which was grown in a standard medium with glucose as well as with crystalline cellulose as the only carbon source. We hypothesized that the inoculum prepared in the presence of Avicel would contain sufficient activities of hydrolases to start hydrolysis of polysaccharides immediately after inoculation. Moreover, this inoculum may contain oligosaccharides required to induce the synthesis of target enzymes. 

Many research groups have used a wide range of agro-industrial lignocellulosic materials as inexpensive raw materials for the production of cellulases and have shown that the production of polysaccharide-hydrolyzing enzymes in WRB is influenced by the type of lignocellulosic substrate [7,8,17,21]. Therefore, besides Avicel, several cheap chemically different lignocellulosic materials were tested in this study as *I. lacteus* growth substrates and stimulate enzyme production. The results in Table 2 show that, regardless of the conditions of preparation of the inoculum, the highest CMCase, xylanase, and β-glucosidase activities were found when the fungus was cultivated in a medium with crystalline cellulose. Wheat straw followed by wheat bran also supported efficient secretion of these enzymes by *I. lacteus*. By contrast, beech sawdust appeared to be a poor substrate for the secretion of three tested enzymes. The most important finding is that replacing glucose with Avicel as a carbon source in the seed culture medium increased the activity of all enzymes in the presence of all growth substrates tested. For example, the CMCase activity of *I. lacteus* in the Avicel-containing medium increased by 38%, xylanase by 50%, and β-glucosidase by 72%. Moreover, the use of inoculum grown in the presence of crystalline cellulose accelerated the production of target enzymes and the peaks of CMCase, and in some cases, xylanase activities were reached much earlier than when using mycelium grown on glucose. 

### 3.4. Scaled Up Enzyme Production in I. lacteus Cultivation in a Fermenter

At the final stage, the possibility of obtaining cellulase and xylanase with a high yield was tested when cultivating the fungus in a fermenter. In the fermentation process, the medium pH was controlled at 6.0 for three days to prevent a decrease of medium pH due to the metabolism of glycerol and create optimal conditions for the rapid development of fungal culture. Then the pH of the medium was controlled at 5.0 to ensure optimal conditions for hydrolysis of cellulose, while after six days, when the culture entered the stationary growth phase, at 5.7, to still allow saccharification of cellulose, but prevent the accumulation of sugars and catabolic repression of enzyme synthesis by *I. lacteus*. 

Traces of cellulase and xylanase activity were detected after two days of fungus cultivation (Figure 2). Thereafter, intense secretion of polysaccharide-hydrolyzing enzymes was observed. The activity of endoglucanase and xylanase reached its maximum after 8 days of fermentation, while FPA was picked on the seventh day of *I. lacteus* cultivation. The enzyme preparation (210 mL) isolated from the culture liquid contained 1710 U/mL CMCase, 2090 U/mL xylanase, and 150 U/mL FPA.

### 3.5. Saccharification of Pretreated Wheat Straw with an Enzyme Preparation from I. lacteus 

In this study, because of the recalcitrant structure, wheat straw (WS) was pretreated with 1.5% NaOH before enzymatic hydrolysis to make the polysaccharide more accessible to the enzymes. The hydrolysis of pretreated WS was compared using a commercial enzyme preparation and the crude enzyme obtained after *I. lacteus* cultivation in the fermenter.

At a FPA load of 20 U/g substrate, the saccharification of 10 mg WS/mL for 3, 6, and 24 h resulted in a reducing sugar yield of 1.2, 2.4, and 4.1 mg/mL, respectively (Figure 3). These concentrations represent yields from wheat straw hydrolysis of 14.5, 29.0, and 49.4% from the theoretically possible, respectively. Doubling the concentration of the WS resulted in an increase of reducing sugars to 1.9, 3.9, 6.2 mg/mL, respectively, but decreased their yield, respectively, to 11.4, 23.5, 37.3% from the theoretically possible. Finally, saccharification of 40 mg WS/mL for 3, 6, and 24 h led to an increase of reducing sugar content up to 3.5, 7.2, and 10.7 mg/mL, respectively. In this case, the yields of reducing sugars were 10.5, 21.7, 32.2% from the theoretically possible. Thus, the degree of hydrolysis of WS and sugar yield was dependent on the substrate concentration and incubation duration. As usual, the rate of cellulose hydrolysis first 6 h was high and decreased over time. One of the reasons may be the accumulation of excess end products, cellobiose and glucose, which inhibit the action of enzymes. Another reason is the partial inactivation of enzymes since enzyme preparations of *I. lacteus* after 24 h of incubation lost an average of 41% of the initial activity, while the commercial preparation lost 38%. Interestingly, the enzyme preparation derived from *I. lacteus* showed a slightly higher cellulose hydrolysis potential than the commercial enzyme. This can be attributed to a more balanced mixture of enzymes and their synergistic interactions [3] as well as to a higher β-glucosidase content, which reduced the inhibitory effect of forming cellobiose. It is worth noting that recently, Mezule and Civzele [19] performed biomass hydrolysis using enzyme preparation obtained from the same strain *I. lacteus* but reaction mixtures containing 3% *w/v* of dry biomass (hay, wood or sawing residue chips, barley straw) and 0.2–0.3 FPU/mL were incubated at pH 5.5 and 30 °C. The highest conversion yields were obtained from hay substrate, where more than 20% of the dry matter of biomass has been converted to fermentable sugar within 24 h of incubation. Barley straw yielded more than 0.1 g fermentable sugar from each g of dry substrate.

## 4. Conclusions

Thus, *I. lacteus* is an excellent producer of polysaccharide-hydrolyzing enzymes, which can be used for saccharification of plant biomass with a high yield of sugars. The cellulase production system is inducible in the presence of cellulose-rich substrates but is suppressed in the presence of an excess of easily metabolizable carbon sources. Consequently, when growing fungus in a fermenter, it is necessary to create conditions that prevent the accumulation of reducing sugars in the medium. Further, the expression of cellulases and xylanases is highly dependent on the type of lignocellulosic growth substrate. The additive effect on the secretion of cellulase and xylanase of *I. lacteus* observed when crystalline cellulose is combined with mandarin pomace may be a good approach to increase the yield of the target enzyme. Our data show that endoglucanase and xylanase are coordinately expressed under all the conditions studied indicating that both enzymes synthesis is under a common regulatory control mechanism. Overall, this study indicates that comprehensive physiological studies are needed to improve the methods of basidiomycetes cultivation, to understand their nutritional requirements and the mechanisms of regulation of hydrolases synthesis, providing the maximum yield of target enzymes.

## Figures and Tables

**Figure 1 biomolecules-11-01341-f001:**
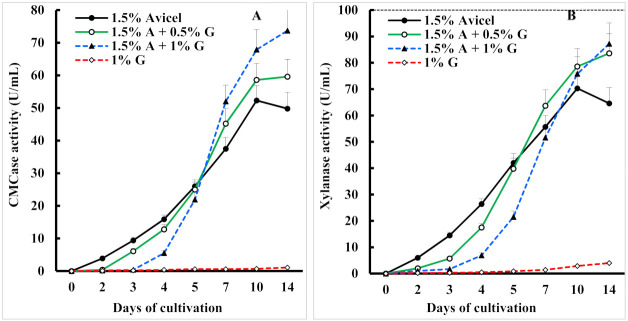
Effect of glycerol (G) as an additional carbon source on the production of endoglucanase (**A**) and xylanase (**B**) during submerged fermentation of Avicel by *I. lacteus*.

**Figure 2 biomolecules-11-01341-f002:**
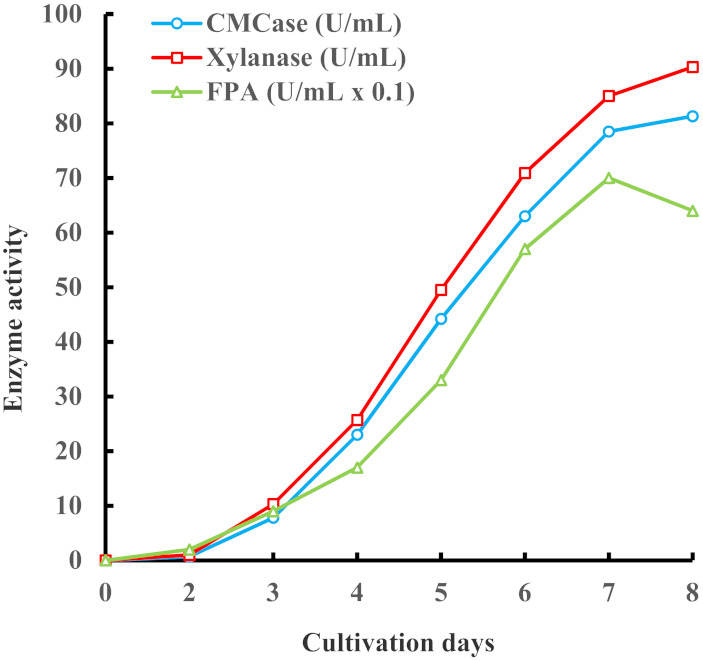
Kinetics of CMCase, xylanase, and FPA accumulation in *I. lacteus* cultivation in fermenter.

**Figure 3 biomolecules-11-01341-f003:**
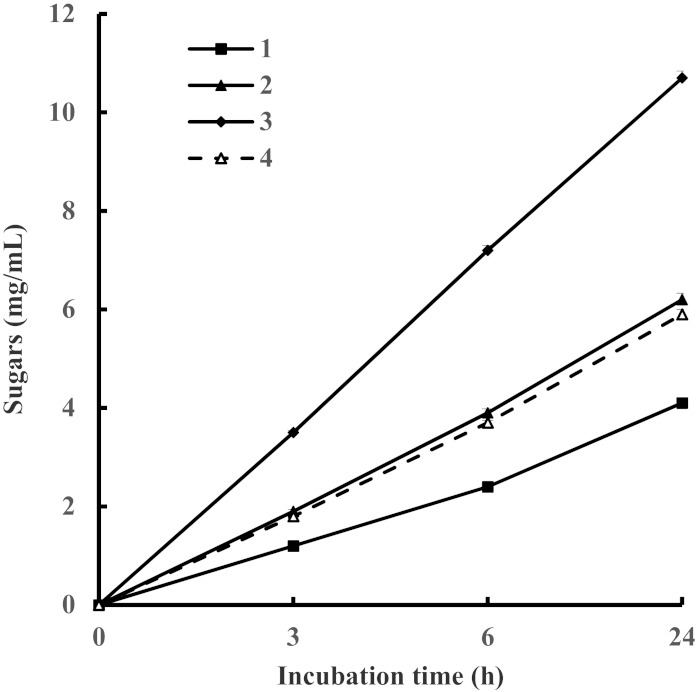
Accumulation of reducing sugars during hydrolysis of pretreated wheat straw (WS) using *I. lacteus* (1–3) and commercial (4) enzyme preparations. Enzyme load–20 U/g substrate. 1–10 mg WS; 2, 4–20 mg WS; 3–40 mg WS.

**Table 1 biomolecules-11-01341-t001:** Modulation of basidiomycetes hydrolytic enzyme activity by chemically different growth substrates.

Growth Substrate	Final pH	CMCase (U/mL)	Xylanase (U/mL)	FPA (U/mL)	β-Glucosidase (U/mL)	β-Xylosidase (U/mL)
*P* *. coccineus*						
1.5% Avicel	5.9 ± 0.1	63 ± 6.0^11^ *	31 ± 3.8^14^ *	4.6 ± 0.5^14^ *	3.6 ± 0.3^11^ *	0.06 ± 0.01^11^ *
4% mandarin squeeze (MS)	6.7 ± 0.1	19 ± 2.1^11^	14 ± 1.1^8^	2.2 ± 0.2^8^	2.2 ± 0.2^8^	0.05 ± 0^6^
4% wheat straw (WS)	6.2 ± 0.1	22 ± 1.4^11^	18 ± 1.2^11^	1.9 ± 0.2^8^	1.5 ± 0.2^6^	0.04 ± 0^6^
1.5% Avicel + 4% MS	5.6 ± 0.1	82 ± 8.3^14^	65 ± 5.8^11^	5.7 ± 0.6^11^	3.3 ± 0.3^8^	0.12 ± 0.01^11^
4% WS +4% MS	5.9 ± 0.1	27 ± 2.4^11^	19 ± 1.7^11^	2.2 ± 0.3^11^	1.6 ± 0.2^6^	0.05 ± 0.01^6^
*S. commune*						
1.5% Avicel	6.1 ± 0.1	39 ± 4.3^14^	626 ± 78^14^	2.1 ± 0.3^11^	10.7 ± 0.8^14^	0.12 ± 0.01^14^
4% MS	6.5 ± 0.1	23 ± 3.0^11^	531 ± 96^8^	3.0 ± 0.3^11^	10.1 ± 0.9^14^	0.15 ± 0.02^14^
4% WS	6.8 ± 0.1	8 ± 1.0^14^	120 ± 10^14^	1.3 ± 0.1^14^	3.8 ± 0.3^14^	0.07 ± 0.01^14^
1.5% Avicel + 4% MS	6.3 ± 0.1	39 ± 2.9^11^	740 ± 80^14^	4.2 ± 0.4^14^	18.6 ± 2.0^14^	0.31 ± 0.03^14^
4% WS + 4% MS	5.9 ± 0.1	14 ± 1.3^11^	528 ± 59^6^	2.2 ± 0.2^14^	5.9 ± 0.4^14^	0.07 ± 0.01^11^
*I. lacteus*						
1.5% Avicel	5.4 ± 0.1	54 ± 6.6^11^	67 ± 8.1^11^	5.1 ± 0.4^11^	1.4 ± 0.12^11^	0.02 ± 0^6^
4% MS	6.9 ± 0.1	18 ± 1.4^8^	20 ± 1.7^11^	1.9 ± 0.2^8^	1.1 ± 0.12^8^	0.02 ± 0^6^
4% WS	6.2 ± 0	23 ± 2.0^11^	29 ± 1.4^8^	2.3 ± 0.2^11^	1.0 ± 0.11^8^	0.05 ± 0.01^8^
1.5% Avicel + 4% MS	5.9 ± 0.1	76 ± 9.0^11^	106 ± 8.6^14^	6.8 ± 0.5^11^	2.3 ± 0.27^8^	0.08 ± 0.01^8^
4% WS + 4% MS	6.3 ± 0.2	40 ± 4.7^8^	34 ± 2.1^8^	2.5 ± 0.3^8^	1.6 ± 0.19^8^	0.05 ± 0.01^6^

* The numbers indicate the days of the peak activity.

**Table 2 biomolecules-11-01341-t002:** Effect of the carbon source in the inoculum medium on the enzyme activity of *I. lacteus*.

Growth Substrate	Final pH	CMCase (U/mL)	Xylanase (U/mL)	β-Glucosidase (U/mL)
	The inoculum was grown in the presence of glucose			
Avicel	5.4 ± 0.1	52.6 ± 6.1^11^ *	71.0 ± 6.1^11^ *	1.65 ± 0.20^8^ *
Mandarin squeeze	6.9 ± 0.1	11.0 ± 0.9^8^	23.8 ± 2.2^14^	0.87 ± 0.06^8^
Wheat bran	5.5 ± 0.1	17.4 ± 1.6^11^	45.2 ± 5.6^11^	1.18 ± 0.16^8^
Wheat straw	5.7 ± 0.1	29.4 ± 3.9^11^	39.1 ± 3.4^11^	1.34 ± 0.14^8^
Beech sawdust	5.7 ± 0.1	3.8 ± 0.2^11^	10.3 ± 1.2^14^	0.21 ± 0.03^11^
	The inoculum was grown in the presence of Avicel			
Avicel	5.7 ± 0.1	72.6 ± 8.0^6^	106.5 ± 9.4^11^	2.84 ± 0.21^8^
Mandarin squeeze	7.0 ± 0.1	14.2 ± 1.1^6^	24.5 ± 1.9^8^	0.82 ± 0.06^11^
Wheat bran	6.0 ± 0.1	24.5 ± 2.0^11^	67.2 ± 6.0^1^^1^	1.46 ± 0.16^11^
Wheat straw	5.5 ± 0.1	46.5 ± 3.4^6^	63.2 ± 7.1^8^	2.08 ± 0.17^8^
Beech sawdust	6.1 ± 0.1	4.3 ± 0.4^6^	8.2 ± 1.1^8^	0.38 ± 0.03^11^

* The numbers indicate the days of the peak activity.

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
