# Peer review of "The Carbon Source Controls the Secretion and Yield of Polysaccharide-Hydrolyzing Enzymes of Basidiomycetes"

_biomolecules, 2021, doi:10.3390/biom11091341_

Round 1

Reviewer 1 Report

Dear Authors,
The manuscript entitled "The Carbon Source Controls the Secretion and Yield of Poly-saccharide-Hydrolyzing Enzymes of Basidiomycetes" fits well with the current trend of using food industry waste in biotechnological processes. The language of the publication seems to be correct and the conclusions to some extent are correct. However, I have a few comments:

- While reading the introduction to the thesis, I felt a lack of examples of practical biotechnological use of enzymes being the subject of the work. Perhaps it would be worth supplementing the introduction with such examples.

- I also have a question regarding the design of the entire experiment why did you not use glucose as sole carbon source to establish a certain baseline (control) of enzyme levels. If such a control was used, it would be worth including it in the paper.

- Ln: 15 "of I. lacteus and P. coccineus" Please note the use of italics for species names throughout the manuscript.

- Analytical methods, especially FPA, should be described in a little more detail.

- Please explain the unit ln: 98 "1 v / v / min" or show the flow in liters per minute for easier understanding of the method.

- Ln: 160-162 "Nevertheless, the results presented in Table 1, in a number of variants of the experiment, clearly show the synergistic effect of the two substrates on the secretion of enzymes." The effect does not seem to be synergistic, but only additive for all the cases mentioned. Please explain how the synergism was calculated? Is this wording adequate? A rough analysis of the data suggests an additive effect.

- References to tables and figures should be given where the data is cited, which facilitates the analysis of the document.

- What do the superscript values ​​in Tables 1 and 2 represent? This should be explained in the table description.

- Ln: 133-134 "calculated by the Excel program (Microsoft Office 2010 package) and only values ​​of p ≤0.05 were considered as statistically significant." Please describe what kind of statistical analysis was carried out
In order to show the differences between the individual carbon sources for the data in Table 1, 2 and Figures 1 , some statistical analysis (e.g. ANOVA) would be indicated. If a statistical analysis was carried out and there were no statistically significant differences, please also mark it on the graph, in the table or in the description. Additionally, Fig 2 does not have any, error bars or SD. It is advisable to complete this data. In how many repetitions has this analysis been carried out?

- Fig. 1. The legend in the figure is somewhat obscure, including additional explanations of what each letter of the legend means in the figure description would greatly facilitate the analysis of the entire manuscript and obtaining key information without a detailed analysis of the content of the manuscript.

- Fig. 2. Please explain in the figure description what FPA (U / mL / 10) means for greater clarity. I would also ask for more detailed descriptions under the figure.

Kind regards,

Reviewer 2 Report

Manuscript titled “The Carbon Source Controls the Secretion and Yield of Poly- 2

saccharide-Hydrolyzing Enzymes of Basidiomycetes” by Metreveli et al is a nice piece of work. Overall, the manuscript is quite strong and informative. Before considering for formal acceptance, I think the draft must be revised properly. My specific comments for author are given below

Authors comments

1 In abstract the species names are not written scientifically and need to be corrected in abstract as well as throughout the drat.

2 Minor improvement in writing is also required throughout the draft.

Reviewer 3 Report

I made some remarks in the attached manuscript version.

Round 2

Reviewer 1 Report

Dear Authors,
"All experiments" in Ln: 139-140 "All experiments were performed twice using three replicates each time. The results are expressed as the mean ± SD. The mean values, as well as standard deviations, were calculated by the Excel program (Microsoft Office 2010 package)"  suggests that all experiments including that carried out in the fermentor to scale up production (Fig. 2) were performed in several iterations. Please make it clear in the text or in the caption under Fig. 2 that it was a single analysis.
Yours faithfully,

Author Response

The reviewer is absolutely right; we changed the first sentence in subsection 2.3.